# Neural processes mediating contextual influences on human choice behaviour

Francesco Rigoli[1], Karl J. Friston[1] & Raymond J. Dolan[1,2]

Contextual influences on choice are ubiquitous in ecological settings. Current evidence suggests that subjective values are normalized with respect to the distribution of potentially available rewards. However, how this context-sensitivity is realised in the brain remains unknown. To address this, here we examine functional magnetic resonance imaging (fMRI) data during performance of a gambling task where blocks comprise values drawn from one of two different, but partially overlapping, reward distributions or contexts. At the beginning of each block (when information about context is provided), hippocampus is activated and this response is enhanced when contextual influence on choice increases. In addition, response to value in ventral tegmental area/substantia nigra (VTA/SN) shows context-sensitivity, an effect enhanced with an increased contextual influence on choice. Finally, greater response in hippocampus at block start is associated with enhanced context sensitivity in VTA/SN. These findings suggest that context-sensitive choice is driven by a brain circuit involving hippocampus and dopaminergic midbrain.

[1] The Wellcome Trust Centre for Neuroimaging, UCL, 12 Queen Square, London WC1N 3BG, UK. [2] Max Planck UCL Centre for Computational Psychiatry and Ageing Research, London WC1B 5EH, UK. Correspondence and requests for materials should be addressed to F.R. (email: f.rigoli@ucl.ac.uk).

The influence of context on value-based choice is substantial and ubiquitous. A classic example is the framing effect, in which risky options are preferred when choices are framed in terms of losses rather than gains[1]. Recent evidence suggests that an influence of context on choice behaviour arises because subjective values are normalized with respect to the distribution of potentially available rewards[2–7]. As an everyday example, this entails that the very same dish will be evaluated as better in a bad restaurant than would be the case if evaluated in a good restaurant.

Recently, there have been attempts to identify neural mechanisms underlying choice adaptation to context-sensitive reward distributions. One candidate mechanism is suggested by the observation that, in several brain structures, activity elicited by reward adapts to context such that an outcome produces a larger response when the associated reward distribution has lower values[2,6,8–11]. This effect is seen in brain regions involved in processing expected value (EV) and reward prediction error (RPE), including ventral striatum[12,13], ventral tegmental area/substantia nigra (VTA/SN)[6,14], orbitofrontal cortex[12,15–18], amygdala[19] and parietal cortex[20]. We recently reported a direct association between context-sensitive reward adaptation in the brain, specifically in VTA/SN, and choice adaptation[6]. However, fundamental questions about the neural substrates of behavioural adaptation to context remain unanswered.

One unanswered question relates to which aspects of neural adaption mediate choice adaptation. Several models are proposed[2,6,8–11,16,21,22], and two key predictions arise out of these. First, neuronal representations of a reference point (for example, reflected in basal neural firing rates) might change so that a context characterized by small rewards would be linked to a lower reference point, leading to enhanced responses to reward with an associated impact on choice behaviour[11,23,24]. Second, choice adaptation might be mediated by a gain modulation, leading to an enhanced signal-to-noise ratio in response to reward and thereby eliciting a context effect on choice[8,10,21,22,25,26]. Both these (additive and multiplicative) proposals entail a normalization that renders subjective value a function of reward that is scaled relative to alternative outcomes.

Another important question regards the precise brain circuits that represent context for reward information. A candidate region is the hippocampus as there is substantial evidence that this region processes contextual information in several cognitive domains[27–30]. For instance, the hippocampus is implicated in contextual fear processing[31–34], in remembering the spatial context in which an object has been encountered[35,36] and in conditional discrimination tasks where contextual information is critical[37]. In addition, recent studies show the hippocampus is involved in complex aspects of reward processing[38–47].

Here, using functional magnetic resonance imaging (fMRI), we investigated the neural underpinnings of choice adaptation to context-specific reward distributions. Participants were presented with a monetary reward, varying trial-by-trial, and were asked to choose between half the amount for sure and a gamble associated with an equal probability of obtaining either the full amount or zero (Fig. 1a). In this way, the two options had equivalent EV. Trials were arranged in short blocks (five trials each), each associated with one of two subtly different gambling contexts involving specific, but partially overlapping, distributions of EV. In a high-value context, possible EVs were £3, £5 and £7, and in a low-value context they were £1, £3 and £5. At the beginning of each block, a panel delivered information about the context, by showing the average trial amount; that is, £6 corresponding to £3 EV, and £10 corresponding to £5 EV for the low- and high-value context, respectively. We predicted this information would elicit activity in regions representing context-sensitive reward distributions, in particular the hippocampus, and that the magnitude of responses would correlate with the degree of contextual adaptation inferred from choice behaviour.

We also exploited the presence of choices common to both contexts (that is, associated with £3 and £5 choices in both contexts) to probe the link between VTA/SN and choice adaptation, by comparing neural responses to identical choices in a low- and high-value context. An increased activation to these choices in a low-value context (and a correlation of this increase with the degree of choice adaptation) would suggest that a modulation in reference point underlies choice adaptation (that is, an additive normalization). Conversely, an increased 'difference' in VTA/SN activation between £5 and £3 choices in the low- compared with high-value context (and a correlation of this effect with the degree of choice adaptation) would suggest that an enhanced signal-to-noise ratio in value signalling (as implied by adaptation of neural gain) underlies choice adaptation (that is, a multiplicative normalization). Formally speaking, in terms of experimental design, the reference point (subtractive normalization) hypothesis predicts a mean effect of context, while the modulation (divisive normalization) hypothesis predicts an interaction between choice (£5 versus £3) and context (low versus high). Importantly, both of these (orthogonal) effects would constitute evidence for contextual normalization of subjective value above and beyond evidence for standard EV theory, implicit in the main effect of choice (£5 versus £3).

Consistent with our predictions, we observed that, at the beginning of each block (when information about context is provided), hippocampus is activated and this response is enhanced when contextual influence on choice increases. When examining choices common to both contexts, we found that response to value in VTA/SN shows context-sensitivity consistent with adaptive gain control, an effect enhanced with an increased contextual influence on choice. Finally, we show that greater response in hippocampus at block start is associated with enhanced context sensitivity in VTA/SN. These findings suggest that context-sensitive choice is driven by a brain circuit involving hippocampus and VTA/SN.

## Results

**Behaviour.** Across participants ($n = 30$), average gambling exceeded 50% (mean = 63; s.d. = 14; t(29) = 24.62, $P < 0.001$; two-tailed $P < 0.05$ was used as the significance criterion for behavioural tests). Such overall risk seeking behaviour is consistent with evidence from studies where, similar to our task, small monetary payoffs were used[48]. Given the fixed relationship between the gamble and the certain gain, the only independent measure varying trial-by-trial was the EV, which was equal for both options (sure and gamble options) on each trial. We assessed the impact of this variable in a logistic regression model of gambling probability, finding that participants gambled more with lower EVs (t-test on the slope parameter of the logistic regression: t(29) = − 2.30, $P = 0.03$). There was no correlation between the individual effect of EV (that is, the slope parameter of the logistic regression model) and the average gambling percentage (Fig. 1b; r(30) = 0.06, $P = 0.74$). The latter result replicates previous findings[6,7] and supports the idea of a differentiation between an average gambling propensity and a preference to gamble with large or small EV as determinant of risk choice.

Using a similar paradigm[6,7], we showed a context effect consistent with the idea that the subjective value of a reward is smaller in the high- compared with low-value context. However, in previous studies, the context changed rarely (about every 10 min) rendering it unclear whether an effect of context emerges

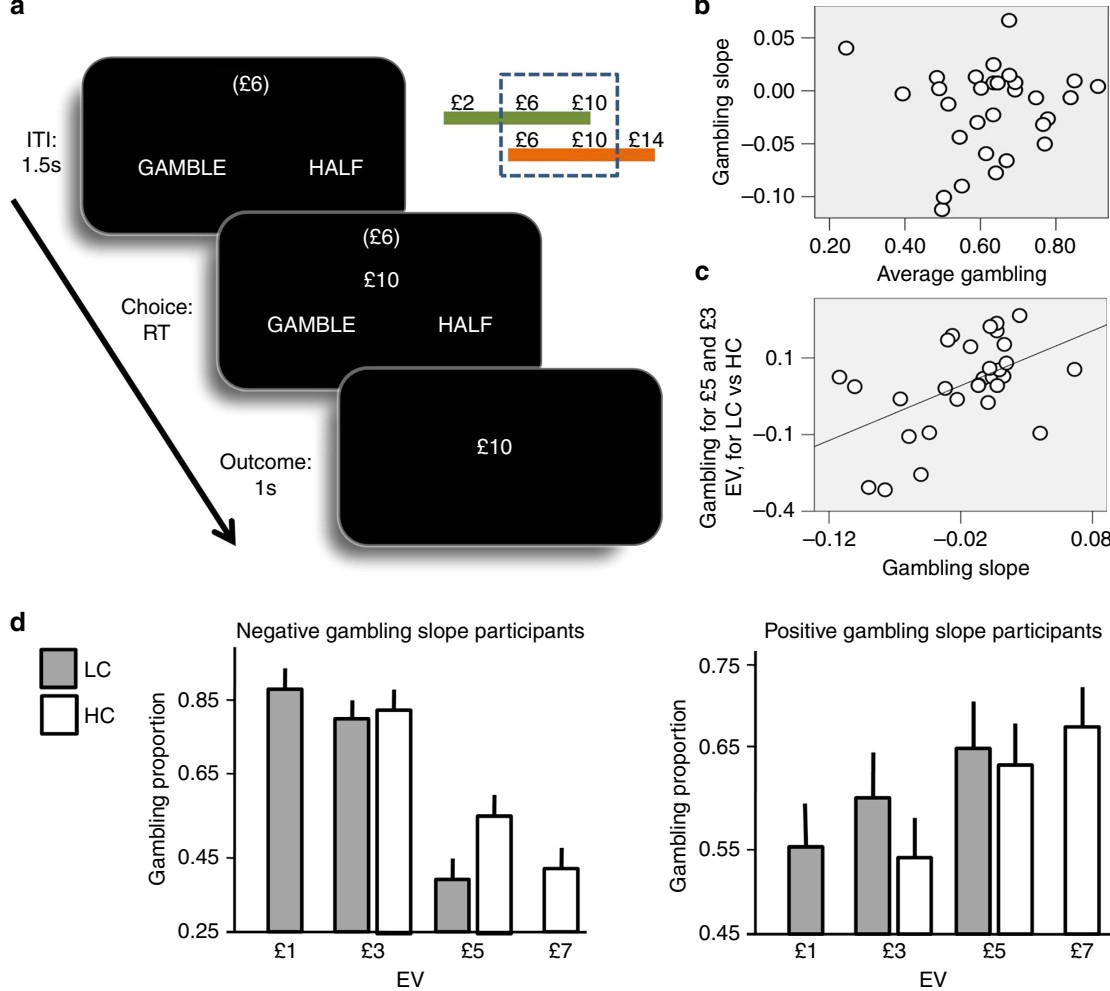

**Figure 1 | Behavioural results.** (**a**) Experimental paradigm: on every trial, participants were presented with a monetary gain amount (£10 in the example) in the centre of the screen. They had to choose between receiving half of it (£5 in the example) for sure or select a 50:50 gamble associated with either the full amount or a zero outcome (hence options had always equivalent EV). After an option was selected (by pressing one of two buttons on a keyboard—left for the gambling, right for the safe option), the outcome appeared for 1 s. During a 1.5 s inter-trial interval, a monetary amount was visible on the top of the screen (in brackets) that indicated the average amount of monetary amount associated with the current block. A low-value context was associated with £2, £6 and £10 amount (corresponding to £1, £3 and £5 EV, respectively), and a high-value context to £6, £10 and £14 amount (corresponding to £3, £5 and £7 EV, respectively). Contexts alternated pseudo-randomly every 5 trials. (**b**) Relationship between individual average gambling proportion (*x*-axis) and the beta weight (labelled as gambling slope; *y*-axis) of the logistic regression of choice behaviour with EV as predictor (r(30) = 0.06, P = 0.74, non significant). (**c**) Relationship between the gambling slope (*x*-axis) and the difference in gambling proportion for £3 and £5 choices (common to both contexts) comparing the low-value context (LC) and the high-value context (HC) (*y*-axis; r(30) = 0.5, P = 0.005). (**d**) Gambling proportion plotted separately for participants with negative (n = 16; on the left) and positive (n = 14; on the right) gambling slope parameter, for different EVs and contexts. Error bars represent standard errors. Considering choices common to both contexts (that is, £3 and £5), it is evident that participants who risked more with decreasing EVs (that is, with a negative gambling slope) gambled more when equivalent choices were smaller compared with the context; whereas participants who risked more with increasing EVs (that is, with a positive gambling slope) gambled more when equivalent choices were larger compared with the context.

only after extensive learning. In the present experiment, we were able to resolve this ambiguity by exploiting a task design where we used short blocks that allowed contexts to alternate quickly (every 30 s).

Consistent with value normalization to context, across participants, we observed a positive correlation between the differential gambling percentage for EVs common to both contexts (that is, the gambling percentage in low-value minus high-value context for £3 and £5), and the effect of EV on gambling percentage (that is, the slope parameter estimated in a logistic regression; Fig. 1c; r(30) = 0.50, P = 0.005). Similar to our previous studies[6,7], this finding shows the direction of a contextual influence depends on a subject-specific propensity to

gamble more with large or small rewards (Fig. 1d). In other words, participants who risked more with increasing EVs gambled more when equivalent choices were larger compared with the context, whereas participants who risked more with decreasing EVs gambled more when equivalent choices were smaller compared with the context. This is consistent with the notion that the subjective value of a reward is smaller in the high- compared with low-value context, and indicates that such contextual effects emerge even without extensive training.

To better characterize the mechanisms underlying choice behaviour and to quantify the level of influence exerted by context on value normalization, we fit a mean-variance return model that computed subjective values consistent with individual

choices. If the reward associated with the safe option was $A$, then the value of the safe option was:

$$V_{\text{CERT}} = A - \chi\tau. \qquad (1)$$

Where, $\chi$ encodes the low- ($\chi = 0$) or high-value context ($\chi = 1$), and the context parameter $\tau$ implements (subtractive) normalization of the reward amount associated with the high-value context. This formulation implies that the mean and variance of the gamble are $A - \chi\tau$ and $(A - \chi\tau)^2$, respectively, making the value of the gamble be:

$$V_{\text{GAMB}} = A - \chi\tau + \alpha(A - \chi\tau)^2 + \mu; \qquad (2)$$

where $\alpha$ is a value-function parameter which determines whether ($\alpha > 0$) or not ($\alpha < 0$) reward variance is attractive, and $\mu$ represents a gambling bias parameter. According to this model, the probability of choosing the gamble is given by a softmax choice rule:

$$\sigma(V_{\text{GAMB}} - V_{\text{CERT}}) = 1/(1 + \exp(-V_{\text{GAMB}} + V_{\text{CERT}})). \qquad (3)$$

We used the Bayesian Information Criterion (BIC; summed across participants) to compare this model with simpler models, where one or two parameters were set to zero: model comparison favoured the full model (model with $\alpha$, $\mu$ and $\tau$, BIC = 19,070; model with $\alpha$ and $\mu$, BIC = 19,427; model with $\alpha$, BIC = 22,866; model with $\mu$, BIC = 22,237).

The value-function parameter $\alpha$ captures a propensity to gamble as a function of reward variance, which in our design corresponds to choice EV. Therefore, we expect this parameter to be correlated with (although not equivalent to) the effect of choice EV on gambling percentage (that is, the effect of EV on gambling percentage as indexed by the slope of a logistic regression), a prediction confirmed by data (r(30) = 0.91, P < 0.001). This ensures that the value-function parameter $\alpha$ has construct validity in relation to (logistic regression) indices of risk preference. An explicit generative model (instead of a logistic regression model) elucidates the computations underlying choice, can be applied to all choices under risk (and not, like the logistic regression model, only when EV is equivalent across options as in our task), and allows estimating the context parameter $\tau$, which is the key variable in our formulation.

To assess whether our model can explain the main behavioural findings, we used the model and subject-specific parameters estimates to generate simulated data and perform the behavioural analyses on the simulated data. Consistent with real data, the full model replicated the lack of correlation between average gambling and the effect of EV on gambling (that is, the slope of the logistic regression) (r(30) = 0.083, P = 0.66), while a correlation emerged when data were simulated using a model without the gambling bias parameter $\mu$ (r(30) = 0.95, P < 0.001). Again consistent with empirical data, the full model replicated the correlation between the effect of EV on gambling and the difference across contexts in gambling for choices common to both contexts (r(30) = 0.54, P = 0.002); a result not obtained using a model without the value-function parameter $\alpha$ (r(30) = 0.14, P = 0.45) or without the context parameter $\tau$ (r(30) = -0.02, P = 0.90).

To examine robustness of model parameters, we estimated new parameters from data simulated with the parameters inferred from real data. Parameters estimated from real data were highly correlated with parameters estimated from simulated data ($\alpha$, r(30) = 0.95, P < 0.001; $\mu$, r(30) = 0.92, P < 0.001; $\tau$, r(30) = 0.89, P < 0.001). Moreover, the average gambling proportion in simulated data was highly correlated with the average gambling proportion in real data (r(30) = 0.92, P < 0.001), and the effect of EV on gambling percentage (that is, the slope parameter of the logistic regression model) in real data was highly correlated with

the same effect in simulated data (r(30) = 0.88, P < 0.001). Collectively, these analyses validate the generative model and show that it can account for the main empirical results.

The generative model allowed us to estimate the degree of context sensitivity as captured by the context parameter $\tau$, so that the relationship between this parameter and neural responses could be investigated. The effect on choice of varying the context parameter is illustrated in Supplementary Fig. 1. As expected, we found that the context parameter $\tau$ was positive across participants (t(29) = 2.7, P = 0.01), indicating that subjective values were normalized so that rewards were afforded less/more subjective values in the high/low context. The context parameter $\tau$ was uncorrelated with other measures (average gambling: r(30) = -0.060, P = 0.750; value-function parameter $\alpha$, r(30) = 0.150, P = 0.430; gambling bias parameter $\mu$, r(30) = 0.120, P = 0.528), ensuring that its relationship with neural responses (reported below) is not confounded by other behavioural factors.

Normalization in the model is subtractive. We compared such model with a model where normalization was divisive, where the value of the sure option is $V_{\text{CERT}} = A/(1 + \chi\tau)$ and the value of the gamble is $V_{\text{GAMB}} = A/(1 + \chi\tau) + \alpha \ (A/(1 + \chi\tau))^2 + \mu$. The divisive normalization version of the model fits less well than the subtractive normalization version (BIC = 19,079 and BIC = 19,070, respectively). The context parameters in the two models were highly correlated (r(30) = 0.87, P < 0.001). To ascertain that the neural results presented below in relation to the parameter $\tau$ were not due to the particular normalization used in the behavioural analysis, we re-ran all the neural analyses using the context parameter extracted from the divisive normalization model, and obtained similar results.

**Imaging**. Our principal goal was to identify the neural correlates of contextual choice adaptation. To do that, we estimated a general linear model (GLM) including a stick function regressor at option presentation separately for each specific EV (£1, £3 and £5 for the low-value context and £3, £5 and £7 for the high-value context) in addition to a stick function regressor at the first trial of blocks. Our ensuing statistical parametric mapping (SPM), analyses focused on regions of interest (ROIs), namely VTA/SN, ventral striatum and hippocampus. For the latter structure, our focus was on the posterior portion, which has been shown to be particularly linked with context processing[27,28,31,32]. ROIs' significance statistics were small-volume corrected (SVC) with P < 0.05 family wise error (see the 'Methods' section for details).

We first tested for brain regions responding to contextual reward information. We reasoned that these regions should be activated following the first trial of each block, when information about context was provided. Hence, we contrasted the regressor associated with the first trial against baseline and found a significant response in right posterior hippocampus (Fig. 2a; 32, -37, -12; Z = 4.02, P = 0.003 SVC; Montreal Neurological Institute coordinates were used) but not VTA/SN or ventral striatum (P > 0.05 SVC). If this signal was linked with processing contextual reward information, one would predict an association between this signal and the impact of context on choice. We tested this both across and within participants. Consistent with our prediction, across individuals we found a positive correlation between the context parameter $\tau$ (reporting the influence exerted by context on choice behaviour) and the response induced by contextual cues (at the start of each block) in the right posterior hippocampus (Fig. 2b; 32, -34, -8; Z = 3.19, P = 0.038 SVC).

When examining variability within single participants, we predicted an increased effect during task sessions characterized by enhanced contextual influence on choice. We tested this by fitting, for each participant, the computational model of

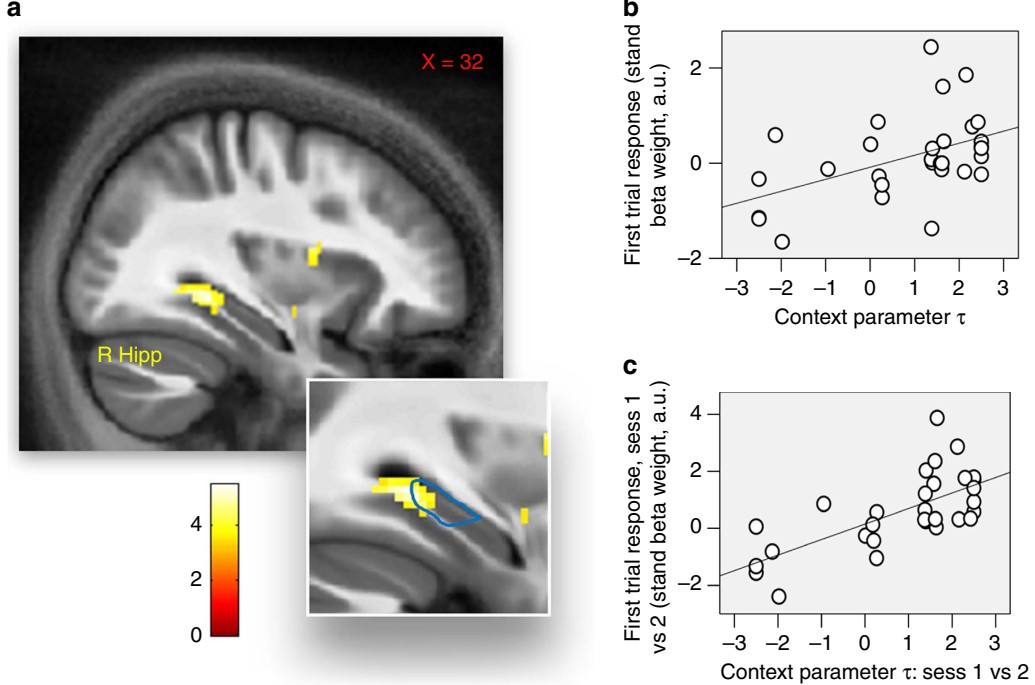

**Figure 2 | Brain response at the first trial of a block.** (**a**) Brain activation in the right hippocampus (Hipp) at first trials of blocks (32, − 37, − 12; $Z = 4.02$, $P = 0.003$ SVC). Significance threshold of $P < 0.005$ is used in the figure for display purposes. The faint blue line represents our ROI relative to posterior hippocampus. (**b**) Relationship between the individual context parameter $\tau$ (reporting, for each participant, the degree of contextual adaptation during the task) and the beta weight relative to first trials of blocks in right hippocampus (32, − 34, − 8; $Z = 3.19$, $P = 0.038$ SVC). Data are plotted for the peak-activation voxel (plot is for display purposes only and no further analyses were performed on these data). (**c**) Relationship between (i) the difference in the individual context parameter $\tau$, when comparing the first and second session of the task and (ii) the beta weight relative to first trials of blocks in right hippocampus, when comparing the first and second session of the task (32, − 29, − 7; $Z = 3.28$, $P = 0.030$ SVC). Data are plotted for the peak-activation voxel (plot is for display purposes only and no further analyses were performed on these data).

behaviour separately for the first and second task session. The two estimates of the context parameter $\tau$ were correlated across subjects ($r(30) = 0.38$, $P = 0.01$) and there was no systematic difference between the first and second session ($t(29) = 1.28$, $P = 0.21$). We considered the difference between the context parameter in the first and second session and investigated the relationship between this difference and neural response to contextual cues. A significant positive correlation was evident in right posterior hippocampus (Fig. 2c; 32, − 29, − 7; $Z = 3.28$, $P = 0.030$ SVC). This analysis shows a within-subjects relationship between responses in hippocampus and the context effect on choice behaviour. Specifically, task sessions associated with increased hippocampal activity for contextual cues were also associated with enhanced contextual adaptation in choice behaviour.

We next investigated context-sensitive (value-related) neural responses and their link with choice adaptation by focusing on the time of option presentation. At this time, activity in VTA/SN and ventral striatum has been shown to correlate with the average EV of options (or with the value of the chosen option)[49]. However, important questions on the role of context remain unanswered. First, it is unclear whether the signal at option presentation adapts to reward distribution expected in a given context. One hypothesis is that adaptation is slow because it depends on an average reward representation, which only changes with extensive experience[23]. Alternatively, the response might adapt immediately to the context, as observed with the presentation of single cues and outcomes[13,14]. Experiments that have manipulated reward context have generally used long blocks (that is, in the order of several minutes)[6], leaving

this issue open. Second, it is unclear whether adaptation can be explained by a change in neural reference point (subtractive normalization) or gain (divisive normalization)[8,10,21,22,25,26]. Third, it is yet to be established whether these putative forms of context-sensitive normalization are related to choice adaptation to context.

To address these questions, we tested two key hypotheses by analysing the response to £3 and £5, choices that were common to both contexts (Fig. 3). A reference point shift predicts increased activation for (common) choices in the low- minus high-value context (a main effect of context). A contextualizing divisive normalization predicts an increased difference in responses to £5 and £3 choices across contexts (an interaction effect; note that a main effect of high- versus low-value choice would identify regions encoding value *per se*).

We first identified areas responding to increasing EV levels, comparing responses at option presentation to the largest EV choice (that is, £7 in the high-value context) with the lowest EV choice (that is, £1 in the low-value context). Increased activity was observed in bilateral ventral striatum (Fig. 4a; left: − 10, 8, − 2; $Z = 5.11$, $P < 0.001$ SVC; right: 12, 13, 0; $Z = 5.21$, $P < 0.001$ SVC) and VTA/SN (Fig. 4b; − 8, − 17, − 15; $Z = 3.80$, $P = 0.005$ SVC). To ensure that the further analyses (reported below) focused on voxels sensitive to EV, activations were masked by a contrast comparing £7 and £1 EV choices, using a $P < 0.005$ uncorrected threshold. For completeness, we also analysed the ventromedial prefrontal cortex, another region involved in processing reward information[50]. Several reports (including our previous study[6]) indicate that, at option presentation, activity in this region reflects the subjective value of the chosen minus the

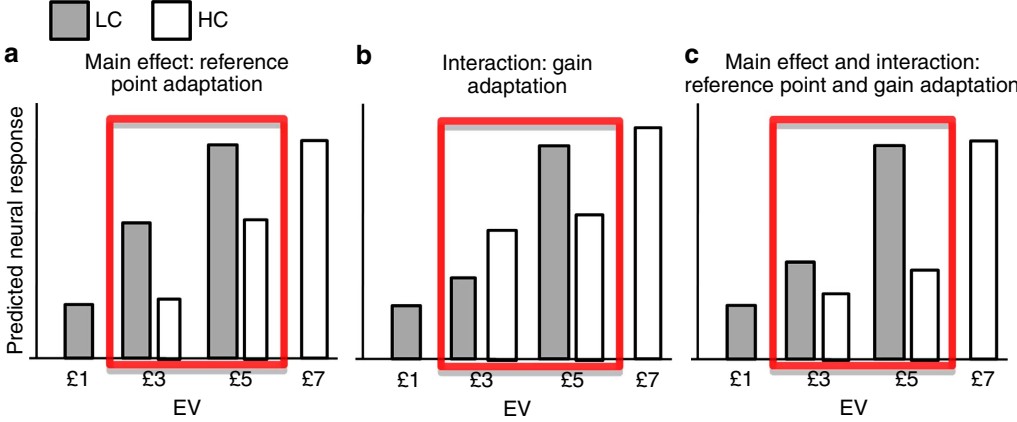

**Figure 3 | Predicted neural response at option presentation in the two contexts for different EVs, as postulated by different hypotheses.** (**a**) According to an hypothesis of a neural reference point which reflects the average contextual EV, a lower reference point is predicted in the low-value context (LC; in grey) compared with the high-value context (HC; in white), leading to larger responses for £3 and £5 EV (highlighted in red) in the former context. Note that the difference in brain activity between £5 and £3 choices is equivalent in the low and high-value context. (**b**) According to the hypothesis that the neural gain increases in low-value contexts, the difference in activation between £5 and £3 choices is larger in the low compared with high-value context. Note that the overall activity when comparing EVs common to both contexts is not necessarily different, as exemplified here. (**c**) Several models of non-linear normalization predict in the low-value context both larger response for EVs common to both contexts and increased difference in activity between £5 and £3 when comparing the two contexts. These predictions are shown here.

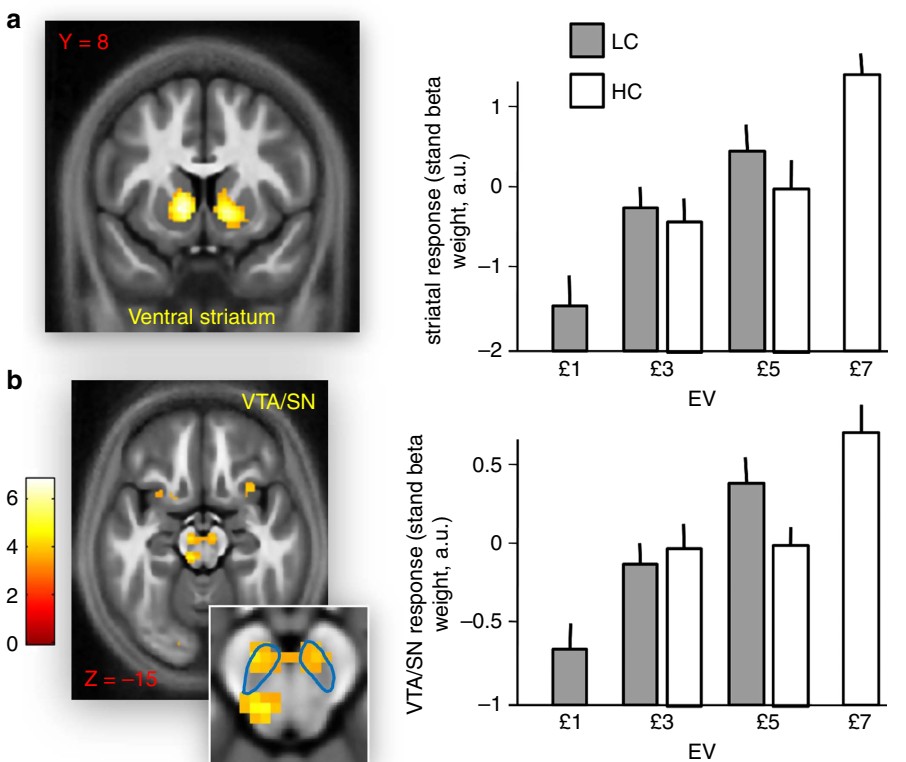

**Figure 4 | Brain response to EV at option presentation.** (**a**) On the left, response in ventral striatum at option presentation for choices associated with £7 EV compared with choices associated with £1 EV (left: −10, 8, −2; $Z = 5.11$, $P < 0.001$ SVC; right: 12, 13, 0; $Z = 5.21$, $P < 0.001$ SVC). Significance threshold of $P < 0.005$ is used in the figure for display purposes. On the right, beta weights (adjusted to each participant's mean) for the choices associated with the different EVs in the low-value context (LC; in grey) and high-value context (HC; in white). Data are shown for the peak-activation voxel from the £7 minus £1 EV contrast. Error lines represent s.e.m. (**b**) On the left, response in VTA/SN at option presentation for choices associated with £7 EV compared with choices associated with £1 EV (−8, −17, −15; $Z = 3.80$, $P = 0.005$ SVC). The faint blue line represents our ROI relative to VTA/SN. On the right, beta weights (adjusted to each participant's mean) for the choices associated with the different EVs and contexts. Data are shown for the peak-activation voxel from the £7 minus £1 EV contrast.

unchosen option and not the average EV of option[50]. As predicted, no voxel in ventromedial prefrontal cortex (defined as a 10 mm sphere ROI centred on prior coordinates[49]: 2, 46, −8)

showed an effect for this contrast (even using $P < 0.05$ uncorrected) and consequently this region was not considered further.

When testing for a main effect of context (subtractive normalization), an increase in the low-value context was seen in bilateral ventral striatum, although only as a trend on the right side (left: $-3$, 10, $-10$; $Z = 3.07$, $P = 0.046$ SVC; right: 4, 10, $-7$; $Z = 2.87$, $P = 0.073$ SVC). Data from VTA/SN were less clear, with a main effect of context in VTA/SN showing only as a weak trend ($-5$, $-19$, $-12$; $Z = 2.49$, $P = 0.095$ SVC; $P = 0.006$ uncorrected). However, an interaction was evident in VTA/SN, indicating that the difference in activity between choices associated with £5 minus choices associated with £3 was larger in the low- compared with the high-value context (Fig. 5a; $-3$, $-24$, $-22$; $Z = 3.23$, $P = 0.036$ SVC). No interaction was detected in ventral striatum ($P > 0.05$ SVC).

We also assessed at a between and within-participants level to investigate whether the context-sensitive effects were related to choice adaptation. We found no evidence in any ROI for a relationship between the context parameter $\tau$ (reporting the influence exerted by context on choice behaviour) and a contrast comparing £5 and £3 choices in the low- minus high-value context. Conversely, across individuals we found a positive correlation between the context parameter $\tau$ and the interaction effect in VTA/SN (Fig. 5b; $-1$, $-22$, $-20$; $Z = 3.22$, $P = 0.037$ SVC). In other words, the change in the differential activation for £5 minus £3 choices across contexts was more pronounced in participants exhibiting a greater contextual influence on choice. As above, we also considered the difference between the context parameter $\tau$ in the first and second session and investigated the relationship between this difference and the neural interaction effect for the first minus second session. A significant correlation was seen in VTA/SN (Fig. 5c; 2, $-22$, $-15$; $Z = 3.39$, $P = 0.018$ SVC).

Overall, these data establish a rapid context-sensitive normalization of subjective value in ventral striatum, consistent with a reference point shift (but not with adaptive gain control), independent of choice adaptation. The data also show a rapid contextual adaptation in VTA/SN consistent with adaptive gain control, with weaker evidence for a reference point shift in this region. Moreover, the adaptive gain control effect in VTA/SN was correlated with choice adaptation.

An intriguing possibility is that the hippocampal encoding of contextual cues mediates a context-sensitive adaptation in VTA/SN, and subsequent choice adaptation. This implicates a modulatory effect, such that adaptation in VTA/SN is enhanced when the hippocampal response to initial trials of blocks (associated with contextual cues) is greater. To test this we performed a psychophysiological interaction analysis (PPI)[51] at the subject level. In PPI analyses, the interaction between a psychological factor and a physiological response is used to predict observed activity elsewhere in the brain. Here, we extracted the individual contrast coefficients reflecting the hippocampal response during initial trials (at the peak-activation voxel). This physiological response was then used to predict the adaptation (that is, interaction comparing low- versus high-value context for £5 minus £3 choices) in VTA/SN. We found a significant PPI in the VTA/SN (7, $-14$, $-12$; $Z = 3.68$, $P = 0.005$ SVC). This result is consistent with an hypothesis that (initial responses in) the hippocampus mediates context-sensitive adaptation in the VTA/SN. In other words, participants with

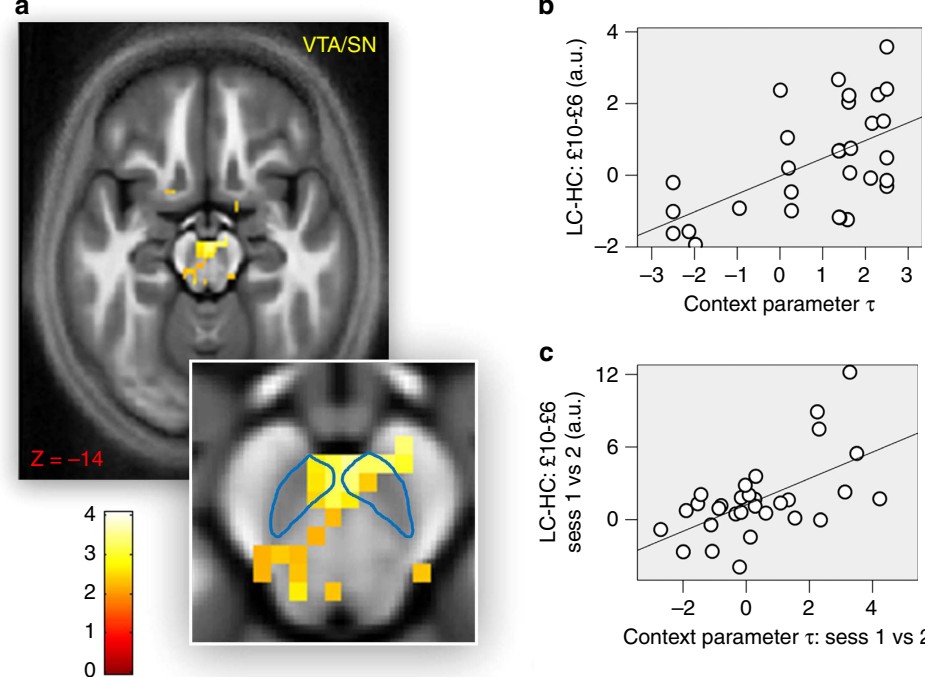

**Figure 5 | The impact of context on activity in VTA/SN.** (**a**) Brain activation at option presentation in VTA/SN for choices associated with £5 EV minus choices associated with £3 EV when comparing low- and high-value context ($-3$, $-24$, $-22$; $Z = 3.23$, $P = 0.036$ SVC). Significance threshold of $P < 0.005$ is used in the figure for display purposes. The faint blue line represents our ROI relative to VTA/SN. (**b**) Relationship between the individual context parameter $\tau$ (reporting, for each participant, the degree of contextual adaptation during the task) and the neural contrast related to choices associated with £5 minus choices associated with £3 when comparing low- and high-value context in VTA/SN ($-1$, $-22$, $-20$; $Z = 3.22$, $P = 0.037$ SVC). Data are plotted for the peak-activation voxel (plot is for display purposes only and no further analyses were performed on these data). (**c**) Relationship between (i) the difference in the individual context parameter $\tau$ when comparing the first and second session of the task and (ii) the contrast related to choices associated with £5 minus choices associated with £3 when comparing low- and high-value context in VTA/SN (2, $-22$, $-15$; $Z = 3.39$, $P = 0.018$ SVC). Data are plotted for the peak-activation voxel (plot is for display purposes only and no further analyses were performed on these data).

increased hippocampal response to contextual cues also exhibited enhanced VTA/SN adaptation. In a subsidiary (within-subjects) analysis, we extracted the individual contrast coefficients reflecting the hippocampal response during initial trials for first versus second task session. This physiological response was then used to predict the difference in adaptation across sessions (that is, the difference across sessions for the interaction comparing low- versus high-value context for £5 minus £3 choices) in VTA/SN. The interaction effect in VTA/SN was again significant ($-4$, $-21$, $-15$; $Z = 3.41$, $P = 0.008$ SVC). This result suggests that sessions with increased hippocampal response to contextual cues were also characterized by enhanced VTA/SN adaptation.

Overall these findings suggest an enhanced neural adaptation in VTA/SN when the hippocampus responds more to contextual cues. This supports an hypothesis that hippocampus is involved in mediating context-sensitive evaluation by controlling response adaptation in VTA/SN, with a subsequent effect on choice adaptation.

## Discussion

Recent evidence suggests contextual effects on choice behaviour are explained by subjective values adapting to the context in which rewards are evaluated, whereby they are increased with a reduction in potentially available rewards[2–7]. Although this effect is relevant in many real-life conditions, its neural underpinnings are largely unknown. Our findings illuminate this context-sensitive evaluation by showing that the hippocampus plays a key role in representing information about reward context, and that its response to contextual information is tightly coupled with the degree of adaptation in choice behaviour.

Across many domains, it is well-established that posterior hippocampus is a key structure in processing contextual cues. For instance, this region is strongly implicated in processing contextual fear[31–34], in remembering the spatial context in which an object has been encountered[35,36], and in conditional discrimination tasks where contextual information is crucial[37]. We build on this evidence, extending the scope of hippocampus to include processing information about reward context as well as highlighting a tight coupling between these contextual influences and choice behaviour. Although fMRI data do not allow causal interpretations, the correlation between neural and behavioural effects (both within and between participants) suggests an hypothesis that responses in this region might endow choice behaviour with context sensitivity.

Recent models of choice behaviour view subjective values as inherently context-dependent, and this is supported by numerous empirical findings[2–7,11,21,22]. This notion is also implicit in models where planning is conceived as (active) inference[52–54], where subjective values become preferences or 'prior beliefs' (about outcomes) that are necessarily normalized, so that they sum to one in any particular context. These models propose that the subjective value of an outcome depends on its relative preference compared with other potential outcomes. In turn, the distribution of potential outcomes is determined by the statistics of the environment, which depend strongly on context. This idea can explain several empirical phenomena including contextual adaptation, the concave shape of the utility function and inter-temporal choice preferences[5].

Our findings are consistent with the idea that hippocampus is crucial for representing a reward context, which in turn influences a computation of subjective values. It has been proposed that subjective values depend on comparing the value of a target outcome against the value of potential outcomes sampled from memory[5]. Within this framework, hippocampal recruitment might

facilitate a sampling of potential outcomes associated with a given context[43]. An alternative possibility is that neural activity in this region represents the sufficient statistics of the context *per se*. These statistics might be used to estimate subjective values, a process which might engage Bayesian approximate inference in the form of free-energy minimization[53–55].

One of the most important finding in rat studies shows that hippocampal neurons are activated when animals occupy specific spatial positions[56,57]. The same neuron can be activated for a location in one spatial context and for another location in another context. However, these neurons tend to respond in corresponding locations in different contexts[58]. For instance, a neuron associated with the centre of an experimental chamber is likely to respond to the centre of a second experimental chamber. It has been hypothesized that this form of neuronal coding might also be deployed in non-spatial tasks[59]. We can speculate that this may also extend to contextual reward processing. One way this could be realised is by different hippocampal cells being preferentially activated by specific instances within a reward context. In addition, similar to reports in the spatial domain[58], this representation might be normalized so that neurons respond to relative values of rewards within the distribution, and hence to different EVs in different contexts, provided their relative subjective value is equivalent. Note, this form of value coding is different from that observed in regions classically linked to value representation, such as ventral striatum and VTA/SN[6,13,14,60–63]. In these regions, neuronal activity correlates with EV (and RPE). Our findings raise a possibility that hippocampal neurons might encode contextual information that is necessary for, or dependent on, a context-sensitive encoding of subjective value.

Recent studies have highlighted a role for hippocampus in (spatial and non-spatial) planning tasks, where value computation is involved, suggesting this structure might be fundamental in representing state-outcome contingencies that are the building blocks of goal-directed or prospective (as opposed to habitual and retrospective) choice[38–47]. This implies that computations more directly related to EV are performed somewhere else in the brain, in structures like basal ganglia, VTA/SN and prefrontal cortex. Our findings raise the possibility that, at least in some cases—for instance, when the context changes rapidly—the hippocampus might play a more direct role in reward processing, and specifically in the contextualization of value.

We note that several studies have shown responses consistent with adaptive coding in ventral striatum[12,13], VTA/SN[6–14], orbitofrontal cortex[12,15–18], amygdala[19] and parietal cortex[20]. For example, in a recent experiment[12], participants chose between variable delayed payment options across two conditions, where the delay spanned either a narrow or wide range. Activation in ventral striatum was consistent with predictions of range adaptation[12]. Here, we extend these findings by showing a link between neural adaptation and choice adaptation, suggesting that the former might mediate the latter[6].

Consistent with previous reports[6], our data indicate that an influence of context on choice behaviour is mediated via VTA/SN neural adaptation at choice presentation. Moreover, in line with previous findings[6], we saw no link between choice and neural adaptation in ventral striatum, supporting further an hypothesis of a distinct role for this structure and VTA/SN in contextual adaptation. Note that a link between VTA/SN and choice adaptation is particularly strong, given that both participants with enhanced choice adaptation exhibited greater interactions between choice and context (that is, neural adaptation) and task sessions with enhanced choice adaptation were characterized by increased neural adaptation. Our data demonstrate that a neural adaptation at option presentation can emerge also when a context changes quickly rendering it unlikely that this process is driven by

a slow accumulation of experience with reward over time[23]. One possibility is that a reward distribution is learnt in association with a context and that this representation is activated when a particular context is presented, and is reflected in activation in VTA/SN and in choice behaviour. More generally, our data support a proposal that normalization processes in the brain might represent a canonical form of neural computation encompassing different cognitive functions, from vision to value-guided choice[25,26].

In our previous study[6], the use of long blocks did not allow us to assess whether VTA/SN adaptation can be explained by a shift in reference point and/or by adaptive gain control. The former hypothesis suggests that, for choices associated with the same EV, responses would increase in a low-value context compared with a high-value context, since the low-value context would be characterized by a smaller reference point. However, our data only marginally support a reference point normalization in VTA/SN (as the corresponding effect emerged only as a weak trend), and they show no relationship between a reference point adaptation and choice adaptation. Conversely, VTA/SN adaptation demonstrated an increased difference between choices associated with £5 and £3 when comparing neural responses in the low- and high-value context, an effect related to choice adaptation.

Several theoretical perspectives suggest that context should induce divisive normalization in both value-related brain regions and choice behaviour[2,8,10,21,22,25,26]. These models are only partially supported by our data, which show that divisive normalization in VTA/SN is linked with an adaptation in choice, but highlight a subtractive—and not divisive—normalization in choice behaviour. This suggests that divisive normalization in VTA/SN may mediate subtractive normalization in choice. However, further theoretical and empirical research is needed to fully understand the link between divisive normalization in VTA/SN and subtractive normalization in choice, and to clarify whether, and how, other aspects of VTA/SN adaptation are involved.

VTA/SN is the main dopaminergic hub in the brain, and substantial evidence supports a central role of dopamine in motivation and adaptive behaviour[23,64,65]. The functions of dopamine are the subject of ongoing debates and one recent proposal has suggested it might be crucial in representing the precision of policies, a concept closely related to neural gain, in the context of incentive value[53]. The observed gain adaptation in VTA/SN linked to choice behaviour seen in our data supports this view, consistent with the idea that this region regulates the incentive value of rewards based on the contextual information, via neural gain control.

An intriguing hypothesis is that the hippocampal response to contextual cues is involved in setting a context by influencing a response adaptation in VTA/SN, and in turn mediating an impact on subsequent choice behaviour. At least three questions arise from this hypothesis. Is there a relationship among contextual effects in hippocampus, VTA/SN and choice behaviour? Do contextual effects in hippocampus precede effects in VTA/SN? Do experimental manipulations of hippocampal response have an impact on contextual effects in VTA/SN response and choice? Our design allowed us to investigate the first two questions. In relation to the first question, we provide evidence of a relationship between contextual influences on hippocampal neural responses and choice, in VTA/SN and choice, and between contextual effects in hippocampus and VTA/SN. In relation to the second question, we found that the context effect in the hippocampus precedes adaptation in VTA/SN, since the former occurs when contextual cues are presented and the latter manifests at option presentation. However, the third question remains open and is likely to require the use of techniques where hippocampal activation can be manipulated directly (for example, through optogenetic interventions).

In conclusion, we provide evidence which suggests that the hippocampus and VTA/SN represent information about the prevailing reward context, and that their responses are associated with the degree of influence of context on choice behaviour. Our results highlight the importance of context in choice and propose a link with its neural substrate. Understanding the cognitive and neural mechanisms of contextual influences is crucial for clarifying the deep nature of choice and to explain important ecological phenomena (for example, in economics) and in psychopathologies (for example, pathological gambling and drug abuse).

## Methods

**Participants.** Thirty healthy right-handed adults (17 females and 13 males, aged 20–40 years, mean age 27 years) participated in the experiment. All participants had normal or corrected-to-normal vision. None had history of head injury, a diagnosis of any neurological or psychiatric condition, or was currently on medication affecting the central nervous system. The study was approved by the University College of London Research Ethics Committee. All participants provided written informed consent and were paid for participating.

**Experimental paradigm and procedure.** During MRI scan, participants performed a computer-based decision-making task lasting ∼40 min (Fig. 1a). On each trial, a monetary amount (referred as trial amount), changing trial-by-trial, was presented in the centre of the screen and participants had to choose whether to accept half of it for sure (pressing a right button) or select a gamble (pressing a left button). The prospects of this choice were always zero and the full monetary amount, each with equal probability. Therefore, on every trial the certain option and the gamble always had the same EV.

The task was organized in short blocks, each comprising five trials. Each block was associated with one of two contexts that determined the possible EVs associated with the block. These EVs were £1, £3 and £5 for the low-value context, and £3, £5 and £7 for the high-value context. Contexts were indicated by the corresponding average trial amount, displayed in brackets on the top of the screen throughout the block, namely £6 and £10 (corresponding to £3 and £5 EV) for the low- and high-value context, respectively. To maximize attention to this contextual cue, the task was made as simple as possible by fixing the buttons used for making choices (that is, the right and left buttons were always used to select the safe option and the gamble, respectively).

Before a new block started, the construction 'New set' appeared for 2 s during the inter-block interval, followed by the context (average trial amount) shown for 2 s. Next, the trial amount of the first trial was displayed followed, right after a response was performed, by the outcome of the choice, shown for 1 s. The block average amount remained on the screen during an inter-trial interval lasting 1.5 s. The order of blocks, trial amounts and outcomes were pseudo-randomized. Participants had 3 s to make their choices; otherwise the statement 'too late' appeared and they received an outcome of zero. At the end of the experiment, one outcome was randomly selected among those received and added to an initial participation payment of £17.

Participants were tested at the Wellcome Trust Centre for Neuroimaging at the University College London. Before scanning, they were fully instructed about the task rules and payment method (that is, they were told that only one outcome would be selected for payment), and practiced for up to 20 unpaid trials. Inside the scanner, participants performed the task in two separate sessions, followed by a 12 min structural scan. After scanning, participants were debriefed and informed about their total remuneration.

**fMRI scanning and analysis.** The task was programmed using the Cogent toolbox (Wellcome Trust Centre for Neuroimaging) in Matlab. Visual stimuli were back projected onto a translucent screen positioned behind the bore of the magnet and viewed via an angled mirror. Blood oxygenation level dependent contrast functional images were acquired with echo-planar T2*-weighted (EPI) imaging using a Siemens Trio 3-Tesla MR system with a 32-channel head coil. To optimize the coverage of our ROIs, a partial volume of the ventral part of the brain was imaged. Each image volume comprised 25 interleaved 3-mm-thick sagittal slices (in-plane resolution = $3 \times 3$ mm; time to echo = 30 ms; repetition time = 1.75 s). The first six volumes were discarded to allow for T1 equilibration effects. T1-weighted structural images were acquired at a $1 \times 1 \times 1$ mm resolution. Functional MRI data were analysed using statistical parametric mapping version 8. Data preprocessing included spatial realignment, unwarping using individual field maps, slice timing correction, normalization and smoothing. Specifically, functional volumes were realigned to the mean volume, were spatially normalized to the standard Montreal Neurological Institute template with a $3 \times 3 \times 3$ voxel size, and were smoothed with 8 mm Gaussian kernel. High-pass filtering with a cut-off of 128 s and AR(1)-model were applied.

Neural responses were modelled with a canonical hemodynamic response function and a GLM including six stimulus functions encoding option presentation separately for each choice EV (£1, £3 and £5 for the low-value context and £3, £5 and £7 for the high-value context). Each of these stick functions was modulated by the corresponding RPE, computed as the difference between the outcomes minus the EV. Thus, RPEs were zero for certain option choices and had positive or negative values when gambles were chosen. The GLM also included a stick function regressor at option presentation for the first trials of each block. This was modulated by a binary variable, indicating whether the block was a high- or low-value context. The GLM was estimated separately for each half of each of the two sessions of the task. The GLM included also 6 movement and 17 physiological (derived from breathing and heart rate signals) nuisance regressors.

Contrasts of interest were computed subject by subject, and used for second-level (between subjects) one-sample $t$-tests and regressions across subjects using standard summary statistic approach[66]. Statistical (SVC) tests focused on the following ROIs: bilateral ventral striatum, VTA/SN and bilateral posterior hippocampus. For VTA/SN we used bilateral anatomical masks manually defined using the software MRIcro and the mean structural image for the group, similar to the approach used in previous studies[67]. For ventral striatum we used a 8 mm sphere centred on *a priori* coordinates extracted from a recent metanalysis[49] (left: $-12$, 12, $-6$; right: 12, 10, $-6$). For posterior hippocampus, we used the template available in the MarsBar Toolbox atlas, and, given our specific interest on the posterior portion, we split the template relative to the vertical axis, resulting in the inclusion of voxels with $z > -12$ coordinates. Statistics of ROIs were SVC using a family wise error rate of $P < 0.05$ as the significance threshold. For exploratory purposes, we also analysed other brain regions where statistics were corrected with respect to the recorded partial volume of the brain, using $P < 0.05$ FWE as the significance threshold. These results are reported in Supplementary Table 1.

**Data availability.** All data necessary to reproduce the results reported are available on request to the corresponding author.

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

## Acknowledgements

This work was supported by the Wellcome Trust (Ray Dolan Senior Investigator Award 098362/Z/12/Z) and the Max Planck Society. The Wellcome Trust Centre for Neuroimaging is supported by core funding from the Wellcome Trust 091593/Z/10/Z. We thank Peter Dayan, Robb Rutledge and Cristina Martinelli for helpful discussions on the topic of the study.

## Author contributions

All authors contributed to designing the study. F.R. performed the study and analysed the data. All authors contributed to discussion and interpretation of the findings and writing the manuscript.

## Additional information

**Competing financial interests:** The authors declare no competing financial interests.

