## [Peer Review File · Nature Communications]

Reviewers' comments:

Reviewer #1 (Remarks to the Author):

Summary

In this study, Rigoli, Friston and Dolan investigated the effect of value context (the mean value in a block of trials) on participants' willingness to gamble and on brain activity associated with value.

The key result is that the posterior hippocampus was active when establishing value context at the start of each value block.

Behaviorally, the degree of contextual influence was shown to correlate with the hippocampal effect.

In the brain, effects of value context on activity in the dopaminergic midbrain (VTA/SN) and ventral striatum were shown as evidence that subjects do change their reward processing due to context. However these could be only rather indirectly linked to behavior and no brain-brain interactions (PPI/DCM) are reported

Evaluation

Although the key result is interesting, much of the rest of the paper is confusing or unconvincing. There quite a bit of modelling, and discussion of divisive vs. subtractive normalization, which seem tangential to the key point. Furthermore the results in this regard contradict each other and the authors' previous work (NeuroImage 2015 - which used a very similar task but couldn't show the hippocampal effect as the blocks were longer), and there is no clear reason why. As a separate point, the modelling section mixes the constructs of risk aversion and gambling slope in a confusing way.

In the end a lot of the material only serves to detract from the key result. I would suggest to cut the lengthy discussions of subtractive and divisive normalizations in favour of a shorter and more focussed paper.

The one thing that would have strengthened the result, in my opinion, would have been to show that the changes in VTA/ventral striatum depended on the hippocampal 'context setting' signal, using PPI or DCM. This would really convince me that the hippocampal signal is setting reward context.

Specifics

I was confused by the results concerning subtractive vs. divisive normalization of value which seemed to contradict each other -

--In the behavioral model (mean-variance model) subtractive normalization is used and this was shown to fit the data better than a model using divisive normalization.

--In the brain, the pattern of value-related activity in ventral striatum was suggestive of subtractive normalization, whilst in the substantia nigra, there was some evidence for divisive rather than subtractive normalization (as the behavioral context effect went up, the activation difference between adjacent reward sizes changed). The authors argue that :

"This is consistent with a divisive normalisation as implied by adaptive gain control, and indicates that

a modulatory process is involved in contextual choice effects, as for instance predicted by influential models postulating a link between divisive normalization and choice adaptation"

--If I understand correctly, the divisive normalization effect in SN/VTA could be linked to behavior, as the relative activation in the two conditions depended on the subtractive normalization parameter τ , which was fit to behavior. In contrast, the subtractive normalization effect (on brain activity) in ventral striatum could not be related to behavior. However, the behavioral data themselves were better fit by a model using subtractive than divisive normalization. In the end I am rather confused which model the authors are arguing for.

Modelling of behavior -

The paper mixed two conceptual constructions between the logistic regression and mean-variance model, which was confusing.

In the original logistic regression analysis, the slope parameter (Fig 1) captures the relationship between stake size (£2-£14) and gambling probability by modelling willingness to gamble as a linear function of expected value (stake size / 2).

But in the mean - variance model, the relationship between stake size and willingness to gamble is captured indirectly, via the variance of the outcome (i.e., parameter α captures whether people are risk-seeking or risk averse, where risk is defined as outcome variance). This depends on the *square* of the expected value (and hence the square of the stake size), not the stake size itself, as in the logistic regression model.

Furthermore, if the probability of winning were to vary rather than being fixed at 0.5, the variance of the outcome would depend on the probability, whilst the slope parameter would not.

In summary the two measures of the individual effect of stake size on gambling probability are not equivalent either mathematically or conceptually. The authors should be clear about which construction they are using.

Note also that commenting would be easier if equations were numbered.

Minor

To aid readability please occasionally re-mention what α and τ are. For example in figure s1 we see the effect of changing α and τ but have to refer back to methods to work out that these are the risk aversion and context normalization parameters respectively.

Figure 1d

Which subjects went into each group? Is this a median split of all subjects or just, say, the lowest 25% and the highest 25%? Please specify in figure legend.

Figure 2

Axis label on part c, and possibly part b, has been accidentally truncated

Figures and 4

The thin blue line is very hard to see

Figure S1

Blue lines referred to sometimes as "green"

Reviewer #2 (Remarks to the Author):

The paper examines context adaptation in risky choice. In each trial of an fMRI experiment subjects chose between a certain amount and a 50-50 gamble of the same expected value (EV). Trials were included in blocks of either high context (HC), in which trials had EV of \$3, \$5 or \$7, or low context (LC), in which trials had EV of \$1, \$3 or \$5. Context was indicated before each block. The authors report that hippocampal activity in response to the context cue was correlated with the degree to which subjects were influenced by context. They then focus on the critical trials of \$3 and \$5, which allow comparison of behavior and neural patterns for the exact same trials in different contexts. The authors report a context adaptation effect in the ventral tegmental area / substantia nigra that is more consistent with gain adaptation than with reference adaptation.

The question of whether and how value representations are normalized is important and timely. The design is simple and straightforward, and the results are potentially interesting. The authors examine explicit context information, unlike other studies that are based on learned context. They also provide nice predictions of what they expect from different normalization models and relate the behavioral and neural results. There are, however, aspects of the analysis that were not clear to me or useful analyses that are missing, and it is also not completely clear how these results are novel compared to some recent results, as detailed below.

- I wasn't sure what the exact model used in the GLM analysis was and how exactly the contrasts were defined. The authors describe separate stick function regressors at option presentation for each choice EV. They also had regressors modulated by reward prediction error (RPE; I assume these were in addition to delta-function predictors, although this is not clear). Why is RPE modeled here? And which predictors are used in the analysis of the responses to \$3 and \$5 in different contexts? It makes sense to compare coefficients of the stick delta functions, because the size of these coefficients will be directly related to value. The coefficients of the RPE predictors, on the other hand, represent the strength of the correlation between the activity and RPE.
- What happens in other value-related areas, especially the vmPFC? Even if the vmPFC does not come up in a whole-brain analysis, ROI analysis will be helpful. Also, the authors should discuss the difference between VTA/SN and ventral striatum in their study.
- A recent paper by Cox and Kable (J Neuroscience 2014) also examined context adaptation, although in a different setting (intertemporal choice). This paper should be discussed, and the novel aspects of the current study should be highlighted.
- One such novel aspect is the hippocampus finding, which is potentially very interesting. It was difficult, however, to evaluate the strength of this finding. The authors focus on an ROI, but if I understand correctly instead of examining the mean activation in the ROI they conduct a voxel-by-voxel analysis and correct for small volume. They then present the peak-activation voxel (Fig 2B) - this is not very convincing. Why not show a scatter plot from the entire region of interest? The same argument goes for the value results (Fig. 4).
- Also, if the hypothesis is that the hippocampus provides context information to value-related structures, it will be helpful to look at connectivity between the hippocampus and these structures. For example, is the degree of connectivity associated with the strength of the context effect across subjects? Is it associated with the degree of context effect across different blocks/parts of the experiment within subject?

Minor comments

- All subjects press left for the sure amount and right for the gamble - why wasn't this counterbalanced across subjects?
- What was the inter-block-interval? Was it the same as inter-trial-interval? If so, why was no jitter introduced with such short intervals?
- Did subjects know in advance that only one trial will be paid? If so, the risk seeking exhibited is a bit surprising, although it may be due to the small amounts overall.

Reviewers' comments:

Reviewer #1 (Remarks to the Author):

Summary

In this study, Rigoli, Friston and Dolan investigated the effect of value context (the mean value in a block of trials) on participants' willingness to gamble and on brain activity associated with value.

The key result is that the posterior hippocampus was active when establishing value context at the start of each value block.

Behaviorally, the degree of contextual influence was shown to correlate with the hippocampal effect.

In the brain, effects of value context on activity in the dopaminergic midbrain (VTA/SN) and ventral striatum were shown as evidence that subjects do change their reward processing due to context. However these could be only rather indirectly linked to behavior and no brain-brain interactions (PPI/DCM) are reported.

We are grateful for the suggestion of a PPI analysis. This analysis is now reported and we believe it substantially strengthens the study. We have now analysed the interaction between context-related activity in VTA and hippocampus, and results are consistent with the hypothesis that hippocampus sets the context and affects adaptation in VTA/SN, as suggested by the reviewers.

We now say (p16, last paragraph):

“An intriguing possibility is that the hippocampal encoding of contextual cues mediates a context sensitive adaptation in VTA/SN, and subsequent choice adaptation. This hypothesis implicates a modulatory effect, such that adaptation in VTA/SN is enhanced when the hippocampal response to the initial trials in a block (associated with contextual cues) is greater. To test this hypothesis we performed a form of Psychophysiological interaction analysis (PPI; Friston et al., 1997) at the subject level. Usually, in PPI analyses, the interaction between a psychological factor and a physiological response is used to predict observed activity elsewhere in the brain. Here, we used a variant of a PPI analysis in which we looked for correlations between the psychophysiological interaction and hippocampal responses during initial trials by switching the explanatory and response variables. In other words, we extracted the individual contrast coefficients reflecting the hippocampal response during initial trials (at the peak-activation voxel). This physiological response was then used to predict the adaptation (i.e., interaction comparing low vs. high-value context for £5 minus £3 choices) in VTA/SN. Using this approach we found a significant PPI in the VTA/SN (7, -14, -12; $Z = 3.68$, $p = 0.005$ SVC). This result is consistent with an hypothesis that (initial responses in) the hippocampus mediates context-sensitive adaptation in the VTA/SN. In other words, participants with increased hippocampal response to contextual cues also exhibited enhanced VTA/SN adaptation. In a subsidiary (within subjects) analysis, we extracted the individual contrast coefficients reflecting the hippocampal response during initial trials for first vs second task session. This physiological response was then used to predict the difference in adaptation across sessions (i.e., the difference across sessions for the interaction comparing low vs. high-value context for £5 minus £3 choices) in VTA/SN.

The interaction effect in VTA/SN was again significant (-4, -21, -15; $Z = 3.41$, $p = 0.008$ SVC). This result suggests that sessions with increased hippocampal response to contextual cues were also characterized by enhanced VTA/SN adaptation, over sessions and subjects.

Overall these findings suggest an enhanced neural adaptation in VTA/SN when the hippocampus responds more to contextual cues. This supports an hypothesis that hippocampus is involved in mediating context sensitive evaluation by controlling response adaptation in VTA/SN, with a subsequent effect on choice adaptation.”

And in the Discussion (p23, last paragraph):

“An intriguing hypothesis is that the hippocampal response to contextual cues is involved in setting a context by influencing a response adaptation in VTA/SN, and in turn mediating an impact on subsequent choice behaviour. This hypothesis can be addressed from three perspectives: (i) is there a relationship among contextual effects in hippocampus, VTA/SN and choice behaviour? (ii) Do contextual effects in hippocampus precede effects in VTA/SN? (iii) Do experimental manipulations of hippocampal response have an impact on contextual effects in VTA/SN response and choice? Our design allowed us to investigate the first two questions. In relation to the first question, we provide evidence of a relationship between contextual influences on hippocampal neural responses and choice, in VTA/SN and choice, and between contextual effects in hippocampus and VTA/SN. In relation to the second question, we found that the context effect in the hippocampus precedes adaptation in VTA/SN, since the former occurs when contextual cues are presented and the latter manifests at option presentation. However, the third question remains open and is likely to require the use of techniques where hippocampal activation can be manipulated directly (e.g., through optogenetic interventions).”

Evaluation

Although the key result is interesting, much of the rest of the paper is confusing or unconvincing. There quite a bit of modelling, and discussion of divisive vs. subtractive normalization, which seem tangential to the key point. Furthermore the results in this regard contradict each other and the authors' previous work (NeuroImage 2015 - which used a very similar task but couldn't show the hippocampal effect as the blocks were longer), and there is no clear reason why.

We thank the reviewer for highlighting this point. We have clarified the discussion of divisive vs. subtractive normalization, and we hope now this is clear.

The reviewer's interpretation is correct (see reviewer's comment below on the same point):

(i) choice was consistent with subtractive normalization; (ii) ventral striatal activity was consistent with subtractive normalization, but there was no relationship between normalization in choice and activity in the ventral striatum; (iii) VTA/SN activity was consistent with divisive normalization, and there was a relationship between subtractive normalization in choice and divisive normalization in VTA/SN.

We agree with the reviewer that these results need to be discussed more thoroughly, and we now do so in the new version. First, we now link these data with models on the impact of neural divisive normalization on choice. Second, we argue that our data suggest that divisive normalization in VTA/SN mediates subtractive normalization in choice. Third, we now explicitly acknowledge that further theoretical and empirical research is needed to fully understand the link between divisive normalization in VTA/SN activity and subtractive normalization in choice. This is presented in the discussion (p22, last paragraph):

“Several theoretical perspectives suggest that context should induce divisive normalization in both value-related brain regions and choice behaviour (Carandini & Heeger, 2012; Cheadle et al., 2014; Louie & Glimcher, 2012; Rangel & Clithero, 2012; Soltani et al., 2012; Summerfield & Tsetsos, 2015). These models are only partially supported by our data, which show that divisive normalization in VTA/SN is linked with an adaptation in choice, but highlight a subtractive – and not divisive - normalization in choice behaviour. This suggests that divisive normalization in VTA/SN may mediate subtractive normalization in choice. However, further theoretical and empirical research is needed to fully understand the link between divisive normalization in VTA/SN and subtractive normalization in choice, and to clarify whether, and how, other aspects of VTA/SN adaptation are involved.”

We would also like to clarify how our results extend our previous findings and do not contradict them (Rigoli et al., Neuroimage 2016). In our previous study, long blocks were adopted in the task (lasting about ten minutes each), implying only four blocks in total. Therefore, the small number of blocks did not allow us to analyse activation in hippocampus at first trials of blocks (in other words we did not have enough power to probe sensitivity to context in this region). The use of multiple short blocks allowed us to test for the role of hippocampus in responding to contextual cues at the start of blocks. We would like to stress that our results on hippocampus do not contradict results from our previous study, since in the previous study the role of the hippocampus was not investigated (nor did we have the required experimental sensitivity).

In addition, in the previous study we observed that, in people showing larger choice adaptation, VTA/SN activity decreased more in the high value context. However, this result can be potentially explained by both subtractive and divisive normalization, a question we

were unable to investigate in that dataset, again due to the long blocks used. Such question could be investigated in the new study (again, thanks to the use of short blocks) as discussed here (p21, last paragraph):

“Consistent with previous reports (Rigoli et al., 2016), our data indicate that an influence of context on choice behaviour is mediated via VTA/SN neural adaptation at choice presentation. Moreover, in line with previous findings (Rigoli et al., 2016), we saw no link between choice and neural adaptation in ventral striatum, supporting further an hypothesis of a distinct role for this structure and VTA/SN in contextual adaptation. Note that a link between VTA/SN and choice adaptation is particularly strong given that both (i) participants with enhanced choice adaptation exhibited greater interactions between choice and context (i.e., neural adaptation) and (ii) task sessions with enhanced choice adaptation were characterized by increased neural adaptation. Our data demonstrate that a neural adaptation at option presentation can emerge also when a context changes quickly rendering it unlikely that this process is driven by a slow accumulation of experience with reward over time (Niv et al., 2007). One possibility is that a reward distribution is learnt in association with a context and that this representation is activated when a particular context is presented, and is reflected in activation in VTA/SN and in choice behaviour. More generally, our data support a proposal that normalization processes in the brain might represent a canonical form of neural computation encompassing different cognitive functions, from vision to value-guided choice (Carandini & Heeger, 2012; Cheadle et al., 2014).

In our previous study (Rigoli et al., 2016), the use of long blocks did not allow us to assess whether VTA/SN adaptation can be explained by a shift in reference point and/or by

adaptive gain control. The former hypothesis suggests that, for choices associated with the same expected value, responses would increase in a low-value context compared to a high-value context, since the low-value context would be characterized by a smaller reference point. However, our data only marginally support a reference point normalisation in VTA/SN (as the corresponding effect emerged only as a weak trend), and they show no relationship between a reference point adaptation and choice adaptation. Conversely, VTA/SN adaptation demonstrated an increased difference between choices associated with £5 and £3 when comparing neural responses in the low and high-value context, an effect related to choice adaptation.”

Therefore, we would like to stress that the result of previous study about VTA/SN adaptation do not contradict results of the new study, but the latter extend the former.

As a separate point, the modelling section mixes the constructs of risk aversion and gambling slope in a confusing way.

We now clarify the distinction between risk aversion and gambling slope – and why and how these are used.

The reviewer is correct concerning the interpretation of the relationship between the value function parameter α and the slope parameter of the logistic regression (see reviewer’s comment below on the same point). We first analyze the slope of the logistic regression because it is a direct index of participants’ risk preference, and can be used as a sanity check to ensure that the model (i) replicates the results of the logistic regression, (ii) the estimated value function parameter α is consistent with direct analysis using logistic regression.

However, we highlight several important advantages of using an explicit model of choice (which includes the value function parameter α). Here, we (i) describe the similarities and differences between the model and logistic regression, (ii) ascertain that value function parameter α is consistent with direct analysis using logistic regression and (iii) highlight the advantages of using a model instead of logistic regression (p9, second paragraph):

“The value function parameter α models a propensity to gamble as a function of reward variance, which in our design is highly correlated with choice EV. Therefore, we expect this parameter to be correlated with (though not equivalent to) the effect of choice EV on gambling percentage; (i.e., the effect of EV on gambling percentage as indexed by the slope of a logistic regression), a prediction confirmed by the data ($r(30)=0.91$, $p<0.001$). This confirms that the estimate of the value function parameter α has construct validity in relation to (logistic regression) indices of risk preference. The advantages of using an explicit generative model of choice (with a value function parameter α), instead of logistic regression are: (i) a generative model makes the computations underlying choice formally explicit, (ii) it can be applied to all choice under risk, while the logistic regression model is meaningful only in the context of our task, where the two options have equivalent EV, (iii) it allows estimation of the context parameter τ , which is the key variable in our formulation.”

We also now report a recovery analysis of the model, confirming that the slope of the logistic regression estimated from data simulated with the model is correlated with the slope estimated from real data (p10, second paragraph):

“Finally, we used the model to recover the parameters using simulated data. The parameters used to generate the data were highly correlated with the parameters estimated from the simulated data (α : $r(30) = 0.95$, $p < 0.001$; μ : $r(30) = 0.92$, $p < 0.001$; τ :

$r(30) = 0.89, p < 0.001$). Moreover, the average gambling proportion in the simulated data was highly correlated with the average gambling proportion in real data ($\alpha: r(30) = 0.92, p < 0.001$), and the effect of EV on gambling percentage (i.e., the slope parameter of the logistic regression model) in the real data was highly correlated with the same effect in the simulated data ($r(30) = 0.88, p < 0.001$).”

In the end a lot of the material only serves to detract from the key result. I would suggest to cut the lengthy discussions of subtractive and divisive normalizations in favour of a shorter and more focussed paper.

The one thing that would have strengthened the result, in my opinion, would have been to show that the changes in VTA/ventral striatum depended on the hippocampal 'context setting' signal, using PPI or DCM. This would really convince me that the hippocampal signal is setting reward context.

We are grateful for this suggestion. We have now analysed the interaction between context-related activity in VTA and hippocampus, and results are consistent with the hypothesis that hippocampus sets the context and affects adaptation in VTA/SN, as suggested by the reviewer. We agree that this new result significantly strengthens the paper.

We now say (p16, last paragraph):

“An intriguing possibility is that the hippocampal encoding of contextual cues mediates a context sensitive adaptation in VTA/SN, and subsequent choice adaptation. This hypothesis implicates a modulatory effect, such that adaptation in VTA/SN is enhanced when the hippocampal response to the initial trials in a block (associated with contextual cues) is

greater. To test this hypothesis we performed a form of Psychophysiological interaction analysis (PPI; Friston et al., 1997) at the subject level. Usually, in PPI analyses, the interaction between a psychological factor and a physiological response is used to predict observed activity elsewhere in the brain. Here, we used a variant of a PPI analysis in which we looked for correlations between the psychophysiological interaction and hippocampal responses during initial trials by switching the explanatory and response variables. In other words, we extracted the individual contrast coefficients reflecting the hippocampal response during initial trials (at the peak-activation voxel). This physiological response was then used to predict the adaptation (i.e., interaction comparing low vs. high-value context for £5 minus £3 choices) in VTA/SN. Using this approach we found a significant PPI in the VTA/SN (7, -14, -12; $Z = 3.68$, $p = 0.005$ SVC). This result is consistent with an hypothesis that (initial responses in) the hippocampus mediates context-sensitive adaptation in the VTA/SN. In other words, participants with increased hippocampal response to contextual cues also exhibited enhanced VTA/SN adaptation. In a subsidiary (within subjects) analysis, we extracted the individual contrast coefficients reflecting the hippocampal response during initial trials for first vs second task session. This physiological response was then used to predict the difference in adaptation across sessions (i.e., the difference across sessions for the interaction comparing low vs. high-value context for £5 minus £3 choices) in VTA/SN. The interaction effect in VTA/SN was again significant (-4, -21, -15; $Z = 3.41$, $p = 0.008$ SVC). This result suggests that sessions with increased hippocampal response to contextual cues were also characterized by enhanced VTA/SN adaptation, over sessions and subjects.

Overall these findings suggest an enhanced neural adaptation in VTA/SN when the hippocampus responds more to contextual cues. This supports an hypothesis that

hippocampus is involved in mediating context sensitive evaluation by controlling response adaptation in VTA/SN, with a subsequent effect on choice adaptation.”

And in the Discussion (p23, last paragraph):

“An intriguing hypothesis is that the hippocampal response to contextual cues is involved in setting a context by influencing a response adaptation in VTA/SN, and in turn mediating an impact on subsequent choice behaviour. This hypothesis can be addressed from three perspectives: (i) is there a relationship among contextual effects in hippocampus, VTA/SN and choice behaviour? (ii) Do contextual effects in hippocampus precede effects in VTA/SN? (iii) Do experimental manipulations of hippocampal response have an impact on contextual effects in VTA/SN response and choice? Our design allowed us to investigate the first two questions. In relation to the first question, we provide evidence of a relationship between contextual influences on hippocampal neural responses and choice, in VTA/SN and choice, and between contextual effects in hippocampus and VTA/SN. In relation to the second question, we found that the context effect in the hippocampus precedes adaptation in VTA/SN, since the former occurs when contextual cues are presented and the latter manifests at option presentation. However, the third question remains open and is likely to require the use of techniques where hippocampal activation can be manipulated directly (e.g., through optogenetic interventions).”

Specifics

I was confused by the results concerning subtractive vs. divisive normalization of value which seemed to contradict each other –

--In the behavioral model (mean-variance model) subtractive normalization is used and this was shown to fit the data better than a model using divisive normalization.

--In the brain, the pattern of value-related activity in ventral striatum was suggestive of subtractive normalization, whilst in the substantia nigra, there was some evidence for divisive rather than subtractive normalization (as the behavioral context effect went up, the activation difference between adjacent reward sizes changed). The authors argue that:

"This is consistent with a divisive normalisation as implied by adaptive gain control, and indicates that a modulatory process is involved in contextual choice effects, as for instance predicted by influential models postulating a link between divisive normalization and choice adaptation"

--If I understand correctly, the divisive normalization effect in SN/VTA could be linked to behavior, as the relative activation in the two conditions depended on the subtractive normalization parameter τ , which was fit to behavior. In contrast, the subtractive normalization effect (on brain activity) in ventral striatum could not be related to behavior. However, the behavioral data themselves were better fit by a model using subtractive than divisive normalization. In the end I am rather confused which model the authors are arguing for.

The reviewer's interpretation is correct: (i) choice was consistent with subtractive normalization; (ii) ventral striatal activity was consistent with subtractive normalization, but there was no relationship between normalization in choice and activity in the ventral striatum; (iii) VTA/SN activity was consistent with divisive normalization, and there was a relationship between subtractive normalization in choice and divisive normalization in VTA/SN.

We agree with the reviewer that these results need to be discussed more thoroughly, and we now do so in the new version. First, we now link these data with models on the impact of neural divisive normalization on choice. Second, we argue that our data suggest that divisive normalization in VTA/SN mediates subtractive normalization in choice. Third, we now explicitly acknowledge that further theoretical and empirical research is needed to fully understand the link between divisive normalization in VTA/SN activity and subtractive normalization in choice. This is presented in the discussion (p22, last paragraph):

“Several theoretical perspectives suggest that context should induce divisive normalization in both value-related brain regions and choice behaviour (Carandini & Heeger, 2012; Cheadle et al., 2014; Louie & Glimcher, 2012; Rangel & Clithero, 2012; Soltani et al., 2012; Summerfield & Tsetsos, 2015). These models are only partially supported by our data, which show that divisive normalization in VTA/SN is linked with an adaptation in choice, but highlight a subtractive – and not divisive - normalization in choice behaviour. This suggests that divisive normalization in VTA/SN may mediate subtractive normalization in choice. However, further theoretical and empirical research is needed to fully understand the link between divisive normalization in VTA/SN and subtractive normalization in choice, and to clarify whether, and how, other aspects of VTA/SN adaptation are involved.”

Modelling of behavior –

The paper mixed two conceptual constructions between the logistic regression and mean-variance model, which was confusing.

In the original logistic regression analysis, the slope parameter (Fig 1) captures the relationship between stake size (£2-£14) and gambling probability by modelling willingness to gamble as a linear function of expected value (stake size / 2).

But in the mean - variance model, the relationship between stake size and willingness to gamble is captured indirectly, via the variance of the outcome (i.e., parameter alpha captures whether people are risk-seeking or risk averse, where risk is defined as outcome variance). This depends on the *square* of the expected value (and hence the square of the stake size), not the stake size itself, as in the logistic regression model.

Furthermore, if the probability of winning were to vary rather than being fixed at 0.5, the variance of the outcome would depend on the probability, whilst the slope parameter would not.

In summary the two measures of the individual effect of stake size on gambling probability are not equivalent either mathematically or conceptually. The authors should be clear about which construction they are using.

The reviewer is correct concerning the interpretation of the relationship between the value function parameter α and the slope parameter of the logistic regression. We first analyze the slope of the logistic regression because it is a direct index of participants' risk preference, and can be used as a sanity check to ensure that the model (i) replicates the results of the logistic regression, (ii) the estimated value function parameter α is consistent

with direct analysis using logistic regression. However, we highlight several important advantages of using an explicit model of choice (which includes the value function parameter α). Here, we (i) describe the similarities and differences between the model and logistic regression, (ii) ascertain that value function parameter α is consistent with direct analysis using logistic regression and (iii) highlight the advantages of using a model instead of logistic regression (p9, second paragraph):

“The value function parameter α models a propensity to gamble as a function of reward variance, which in our design is highly correlated with choice EV. Therefore, we expect this parameter to be correlated with (though not equivalent to) the effect of choice EV on gambling percentage; (i.e., the effect of EV on gambling percentage as indexed by the slope of a logistic regression), a prediction confirmed by the data ($r(30)=0.91$, $p<0.001$). This confirms that the estimate of the value function parameter α has construct validity in relation to (logistic regression) indices of risk preference. The advantages of using an explicit generative model of choice (with a value function parameter α), instead of logistic regression are: (i) a generative model makes the computations underlying choice formally explicit, (ii) it can be applied to all choice under risk, while the logistic regression model is meaningful only in the context of our task, where the two options have equivalent EV, (iii) it allows estimation of the context parameter τ , which is the key variable in our formulation.”

We also report a recovery analysis of the model, confirming that the slope of the logistic regression estimated from data simulated with the model is correlated with the slope estimated from real data (p10, second paragraph):

“Finally, we used the model to recover the parameters using simulated data. The parameters used to generate the data were highly correlated with the parameters

estimated from the simulated data (α : $r(30) = 0.95$, $p < 0.001$; μ : $r(30) = 0.92$, $p < 0.001$; τ : $r(30) = 0.89$, $p < 0.001$). Moreover, the average gambling proportion in the simulated data was highly correlated with the average gambling proportion in real data (α : $r(30) = 0.92$, $p < 0.001$), and the effect of EV on gambling percentage (i.e., the slope parameter of the logistic regression model) in the real data was highly correlated with the same effect in the simulated data ($r(30) = 0.88$, $p < 0.001$)."

Note also that commenting would be easier if equations were numbered.

Thank you: the equations are now numbered.

Minor

To aid readability please occasionally re-mention what alpha and tau are. For example in figure s1 we see the effect of changing alpha and tau but have to refer back to methods to work out that these are the risk aversion and context normalization parameters respectively.

We apologise for this. Every time a parameter is mentioned in the text, we now say "value function parameter α ", "gambling bias parameter μ ", and "context parameter τ " respectively.

Figure 1d: Which subjects went into each group? Is this a median split of all subjects or just, say, the lowest 25% and the highest 25%? Please specify in figure legend.

We considered all subjects with a negative gambling slope ($n=16$) in group one, and all subjects with a positive gambling slope ($n=14$) in group two. This is now clarified (fig 1D, caption):

“Gambling proportion was plotted separately for all participants with negative ($n=16$; on the left) and positive ($n=14$; on the right) gambling slope parameter”

Figure 2: Axis label on part c, and possibly part b, has been accidentally truncated

We apologise for this. It is now corrected.

Figures and 4: The thin blue line is very hard to see

We apologise for this. Blue lines for all figures are now thicker.

Figure S1: Blue lines referred to sometimes as "green"

Apologize, this is now corrected.

Reviewer #2 (Remarks to the Author):

The paper examines context adaptation in risky choice. In each trial of an fMRI experiment subjects chose between a certain amount and a 50-50 gamble of the same expected value

(EV). Trials were included in blocks of either high context (HC), in which trials had EV of \$3, \$5 or \$7, or low context (LC), in which trials had EV of \$1, \$3 or \$5. Context was indicated before each block. The authors report that hippocampal activity in response to the context cue was correlated with the degree to which subjects were influenced by context. They then focus on the critical trials of \$3 and \$5, which allow comparison of behavior and neural patterns for the exact same trials in different contexts. The authors report a context adaptation effect in the ventral tegmental area / substantia nigra that is more consistent with gain adaptation than with reference adaptation.

The question of whether and how value representations are normalized is important and timely. The design is simple and straightforward, and the results are potentially interesting. The authors examine explicit context information, unlike other studies that are based on learned context. They also provide nice predictions of what they expect from different normalization models and relate the behavioral and neural results. There are, however, aspects of the analysis that were not clear to me or useful analyses that are missing, and it is also not completely clear how these results are novel compared to some recent results, as detailed below.

- I wasn't sure what the exact model used in the GLM analysis was and how exactly the contrasts were defined. The authors describe separate stick function regressors at option presentation for each choice EV. They also had regressors modulated by reward prediction error (RPE; I assume these were in addition to delta-function predictors, although this is not clear). Why is RPE modeled here? And which predictors are used in the analysis of the responses to \$3 and \$5 in different contexts? It makes sense to compare coefficients of the stick delta functions, because the size of these coefficients will be directly related to value.

The coefficients of the RPE predictors, on the other hand, represent the strength of the correlation between the activity and RPE.

We thank the reviewer for raising this point. The GLM included separate stick stimulus function for each choice EV (i.e., 6 regressors in total), modulated by outcome RPE. RPEs were computed as the difference between the outcomes minus the EV. Thus, RPEs were zero for certain option choices and had positive or negative values for choices of the gamble. In the analysis of the responses to £3 and £5 we considered the stick function regressors (not the RPE parametric modulators). This is now clarified (p27, second paragraph):

“Neural responses were modelled with a canonical hemodynamic response function and a GLM including six stimulus functions encoding option presentation separately for each choice EV (£1, £3, and £5 for the low-value context and £3, £5 and £7 for the high-value context). Each of these stick functions was modulated by the corresponding RPE, computed as the difference between the outcomes minus the EVs. Thus, RPEs were zero for certain option choices and had positive or negative values when gambles were chosen.”

- What happens in other value-related areas, especially the vmPFC? Even if the vmPFC does not come up in a whole-brain analysis, ROI analysis will be helpful. Also, the authors should discuss the difference between VTA/SN and ventral striatum in their study.

This is an interesting question. Please note that, in our previous experiment using a similar paradigm (Rigoli et al., 2016), we found that activation in vmPFC (contrary to VTA/SN and ventral striatum) did not reflect choice EV at option presentation. Rather, vmPFC activity

was more consistent with encoding the subjective value difference between the chosen and unchosen option, which in our design is uncorrelated with choice EV (please see Rigoli et al. (2016) for details). This is also consistent with other findings (e.g., Rushworth et al., 2011), and this is the reason why vmPFC was not initially included here as ROI. Here, our interest was in areas where activity predicted choice EV at option presentation. However, we agree we should explain/mention this explicitly and report data for vmPFC. We now report that consistent with previous evidence (Rigoli et al., 2016; Rushworth et al., 2011), no voxel in vmPFC responded to £7 minus £1 at option presentation, even when using $p < 0.05$ uncorrected as threshold. Therefore, vmPFC was not investigated further. This is now clarified (p14, last paragraph):

“We first identified areas responding to increasing EV levels. We compared responses at option presentation to the largest EV choice (i.e., £7 in the high-value context) with the lowest EV choice (i.e., £1 in the low-value context). Increased activity was observed in bilateral ventral striatum (fig 4A; left: -10, 8, -2; $Z = 5.11$, $p < 0.001$ SVC; right: 12, 13, 0; $Z = 5.21$, $p < 0.001$ SVC) and VTA/SN (fig 4B; -8, -17, -15; $Z = 3.80$, $p = 0.005$ SVC). To ensure that the further analyses (reported below) focused on voxels sensitive to EV, activations were masked by a contrast comparing £7 and £1 EV choices, using a $p < 0.005$ uncorrected threshold. For completeness, we also analysed the ventromedial prefrontal cortex (vmPFC), another region involved in processing reward information (Rushworth et al., 2011). Several reports (including our previous study (Rigoli et al., 2016)) indicate that, at option presentation, activity in this region reflects the subjective value of the chosen minus the unchosen option and not the average EV of option (Rushworth et al., 2011). As predicted, no voxel in vmPFC (defined as a 10mm sphere ROI centred on prior coordinates 2, 46, -8; Bartra

et al., 2013) showed an effect for this contrast (even using $p < 0.05$ uncorrected) and consequently this region was not considered in further analyses”

- A recent paper by Cox and Kable (J Neuroscience 2014) also examined context adaptation, although in a different setting (intertemporal choice). This paper should be discussed, and the novel aspects of the current study should be highlighted.

Thanks for the important reference. We argue that previous literature (e.g., Cox & Kable, 2014) has left open the question of whether neural adaptation has any behavioural effect. Our study addresses this question; since it establishes a link between neural adaptation and choice adaptation. We clarify this in the Discussion (p20, last paragraph):

“We note that several studies have shown responses consistent with adaptive coding in ventral striatum (Cox & Kable, 2014; Park et al., 2012), ventral tegmental area/substantia nigra (VTA/SN; Rigoli et al., 2016; Tobler et al., 2005), orbitofrontal cortex (Cox & Kable, 2014; Kobayashi et al., 2010; Padoa-Schioppa, 2009; Padoa-Schioppa & Assad, 2008; Tremblay & Schultz, 1999), amygdala (Bermudez & Schultz, 2010), and parietal cortex (Louie et al., 2011). For example, in a recent experiment (Cox & Kable, 2014), participants chose between variable delayed payment options across two conditions, where the delay spanned either a narrow or wide range. Activation in ventral striatum was consistent with predictions of range adaptation (Cox & Kable, 2014). Here, we extend these findings by showing a link between neural adaptation and choice adaptation, suggesting that the former might mediate the latter (see also Rigoli et al., 2016).”

- One such novel aspect is the hippocampus finding, which is potentially very interesting. It was difficult, however, to evaluate the strength of this finding. The authors focus on an ROI, but if I understand correctly instead of examining the mean activation in the ROI they conduct a voxel-by-voxel analysis and correct for small volume. They then present the peak-activation voxel (Fig 2B) - this is not very convincing. Why not show a scatter plot from the entire region of interest? The same argument goes for the value results (Fig. 4).

Thanks for pointing this out. We agree with the reviewer that examining the mean activation of ROIs is an alternative which could be considered. However, we would like to stress that conducting a voxel-by-voxel analysis (and correcting for small volume) is equally valid statistically. Furthermore, many people regard the use of ROI averages as inappropriate (because they overlook spatial variations in systematic responses; e.g., centre-surround responses): see: A critique of functional localisers Neuroimage. 2006 May 1;30(4):1077-87. Therefore, we would prefer to remain with our current analyses, which are (from our perspective) standard practice.

We agree that to avoid misunderstanding, we should explicitly write "Plot is for display purposes only and no further analyses were performed on these data" when scatterplots are presented. This statement is now included where relevant.

- Also, if the hypothesis is that the hippocampus provides context information to value-related structures, it will be helpful to look at connectivity between the hippocampus and these structures. For example, is the degree of connectivity associated with the strength of

the context effect across subjects? Is it associated with the degree of context effect across different blocks/parts of the experiment within subject?

We are grateful for this suggestion. We have now analyzed the connection between context-related activity in VTA and hippocampus, and results are consistent with the hypothesis that hippocampus sets the context and affects adaptation in VTA/SN, as suggested by the reviewer. We believe that this new results significantly strengthen the paper.

We now say (p16, last paragraph):

“An intriguing possibility is that the hippocampal encoding of contextual cues mediates a context sensitive adaptation in VTA/SN, and subsequent choice adaptation. This hypothesis implicates a modulatory effect, such that adaptation in VTA/SN is enhanced when the hippocampal response to the initial trials in a block (associated with contextual cues) is greater. To test this hypothesis we performed a form of Psychophysiological interaction analysis (PPI; Friston et al., 1997) at the subject level. Usually, in PPI analyses, the interaction between a psychological factor and a physiological response is used to predict observed activity elsewhere in the brain. Here, we used a variant of a PPI analysis in which we looked for correlations between the psychophysiological interaction and hippocampal responses during initial trials by switching the explanatory and response variables. In other words, we extracted the individual contrast coefficients reflecting the hippocampal response during initial trials (at the peak-activation voxel). This physiological response was then used to predict the adaptation (i.e., interaction comparing low vs. high-value context for £5 minus £3 choices) in VTA/SN. Using this approach we found a significant PPI in the VTA/SN (7, -14, -12; $Z = 3.68$, $p = 0.005$ SVC). This result is consistent with an hypothesis that

(initial responses in) the hippocampus mediates context-sensitive adaptation in the VTA/SN. In other words, participants with increased hippocampal response to contextual cues also exhibited enhanced VTA/SN adaptation. In a subsidiary (within subjects) analysis, we extracted the individual contrast coefficients reflecting the hippocampal response during initial trials for first vs second task session. This physiological response was then used to predict the difference in adaptation across sessions (i.e., the difference across sessions for the interaction comparing low vs. high-value context for £5 minus £3 choices) in VTA/SN. The interaction effect in VTA/SN was again significant (-4, -21, -15; $Z = 3.41$, $p = 0.008$ SVC). This result suggests that sessions with increased hippocampal response to contextual cues were also characterized by enhanced VTA/SN adaptation, over sessions and subjects.

Overall these findings suggest an enhanced neural adaptation in VTA/SN when the hippocampus responds more to contextual cues. This supports an hypothesis that hippocampus is involved in mediating context sensitive evaluation by controlling response adaptation in VTA/SN, with a subsequent effect on choice adaptation.”

And in the Discussion (p23, last paragraph):

“An intriguing hypothesis is that the hippocampal response to contextual cues is involved in setting a context by influencing a response adaptation in VTA/SN, and in turn mediating an impact on subsequent choice behaviour. This hypothesis can be addressed from three perspectives: (i) is there a relationship among contextual effects in hippocampus, VTA/SN and choice behaviour? (ii) Do contextual effects in hippocampus precede effects in VTA/SN? (iii) Do experimental manipulations of hippocampal response have an impact on contextual effects in VTA/SN response and choice? Our design allowed us to investigate the first two questions. In relation to the first question, we provide evidence of a relationship between

contextual influences on hippocampal neural responses and choice, in VTA/SN and choice, and between contextual effects in hippocampus and VTA/SN. In relation to the second question, we found that the context effect in the hippocampus precedes adaptation in VTA/SN, since the former occurs when contextual cues are presented and the latter manifests at option presentation. However, the third question remains open and is likely to require the use of techniques where hippocampal activation can be manipulated directly (e.g., through optogenetic interventions).”

Minor comments

- All subjects press left for the sure amount and right for the gamble - why wasn't this counterbalanced across subjects?

Thanks for raising this question. In our previous study using a similar paradigm (Rigoli et al., 2016), the button used to choose the safe option varied across trials. However, in the new study, blocks vary much more quickly, rendering contextual information potentially harder to attend. Therefore, we modified the task so to enhance the relevance of the contextual information – by minimizing the relevance of any other factor, including which buttons were required to choose the options. This is now explained in the manuscript (p25, second paragraph):

“The task was organized in short blocks, each comprising 5 trials. Each block was associated with one of two contexts that determined the possible EVs associated with the block. These EVs were £1, £3 and £5 for the low-value context, and £3, £5 and £7 for the high-value context. Contexts were indicated by the corresponding average trial amount, displayed in

brackets on the top of the screen throughout the block, namely £6 and £10 (corresponding to £3 and £5 EV) for the low and high value context respectively. To maximize attention to this contextual cue, the task was made as simple as possible by fixing the buttons used for making choices (i.e., the left and right buttons were always used to select the safe option and the gamble respectively).”

- What was the inter-block-interval? Was it the same as inter-trial-interval? If so, why was no jitter introduced with such short intervals?

The inter-block interval is now reported (p25, last paragraph):

“Before a new block started, the construction “New set” appeared for two seconds during the inter-block interval, followed by the context (average trial amount) shown for two seconds.”

We carefully analysed the design matrix to ensure that the associated regressor estimates were largely uncorrelated. Given that trials and blocks were randomized, regressors were largely uncorrelated. Therefore there was no need for long inter-block or inter-trial intervals and no need for jittering (jittering is only useful if you are modelling delay period activity – or the jitter is substantially greater than the time constants of the dynamic response function. In the absence of any constraints on the inter-block intervals, we used a short interval to provide more trials and hence a more efficient design.

- Did subjects know in advance that only one trial will be paid? If so, the risk seeking exhibited is a bit surprising, although it may be due to the small amounts overall.

Thanks for highlighting this point. We now clarify that subjects knew in advance that only one trial would be paid (p26, second paragraph):

“Before scanning, they were fully instructed about the task rules and payment method (i.e., they were told that only one outcome would be selected for payment), and practiced for up to 20 unpaid trials.”

We agree with the reviewer that the overall risk seeking behaviour is not uncommon in experiments where small monetary amounts are involved, and this has now been clarified (p6, last paragraph):

“Across participants, average gambling exceeded 50% (mean = 63; SD = 14; $t(29) = 24.62$, $p < 0.001$; two-tailed $p < 0.05$ was used as the significance criterion for all behavioural tests). Such overall risk seeking behaviour is consistent with evidence from studies where, similar to our task, small monetary payoffs were used (Prelec & Loewenstein, 1991).”

REVIEWERS' COMMENTS:

Reviewer #1 (Remarks to the Author):

In the response letter and revised manuscript, the authors have addressed most of my points. I am pleased that the functional connectivity analysis has been included as I think this strengthens the paper.

The authors have made some changes to make the paper more readable but it is still at once dense with many analyses, and lacking sufficient signposting. Since this is a general interest journal I think the onus is on the authors to make sure that someone who is not a die-hard neuroeconomist can still see why each result is important and how the analyses hang together to support a certain model of contextual influence on choice.

I think the authors could have made it easier going for the reader by

a) being explicit each time an analysis is introduced, how this bears on the main hypothesis of the paper (this may appear clear to the authors but was not always clear to me). Several analyses are introduced by statements which, whilst true, make no link to the hypotheses or theoretical context of the paper, e.g. to paraphrase, 'we did this modelling to better characterise responses' or 'we used subtractive normalisation but we could have used divisive normalisation, so we checked that too'

b) giving as concise and explicit a statement of the key results as possible, in each of the introduction/discussion/abstract.

Instead of stating 'there was a relationship between x and y' stating the nature of the relationship 'as x increases, y increases'.

For example (but this is a general comment about the whole paper) in the abstract it says:

"When we examined responses to choices common to both contexts, activity in ventral tegmental area/substantia nigra showed a context-sensitivity consistent with adaptive neuronal gain control, an effect correlating with the influence of context on choice, both across and within participants"

why not say:

"Activity in dopaminergic regions (VTA/SN) was context sensitive, such that a reward of a given size evoked a greater response in the context of a low reward block. This effect was stronger in participants who were more prone to gamble in low reward blocks. Within participants, increased HPC activity at the start of each block predicted increased context sensitivity in that block, both behaviourally and in the dopaminergic midbrain signal"

...or some similar explicit statement that reflects your exact conclusions?

Obviously, it is the authors' own work and they should write in their own style. I am only pointing out that I thought a lot of the work of motivating the reader was left up to the reader himself and, had I not been obliged to finish the paper because I was reviewing it, I might have gotten a bit lost in the sea of analyses and given up, which would have been a pity because the results, when stated concisely, are actually quite interesting and important.

Reviewer #2 (Remarks to the Author):

The authors have done a good job in responding to comments. The new PPI analysis substantially strengthens the paper.

REVIEWERS' COMMENTS:

Reviewer #1 (Remarks to the Author):

In the response letter and revised manuscript, the authors have addressed most of my points. I am pleased that the functional connectivity analysis has been included as I think this strengthens the paper.

Many thanks for the very useful comments.

The authors have made some changes to make the paper more readable but it is still at once dense with many analyses, and lacking sufficient signposting. Since this is a general interest journal I think the onus is on the authors to make sure that someone who is not a die-hard neuroeconomist can still see why each result is important and how the analyses hang together to support a certain model of contextual influence on choice.

We are grateful for the feedback on this aspect. We have attempted to clarify the rationale and the results of our analysis, as reported below.

I think the authors could have made it easier going for the reader by

a) being explicit each time an analysis is introduced, how this bears on the main hypothesis of the paper (this may appear clear to the authors but was not always clear to me). Several analyses are introduced by statements which, whilst true, make no link to the hypotheses or theoretical context of the paper, e.g. to paraphrase, 'we did this modelling to better characterise responses' or 'we used subtractive normalisation but we could have used divisive normalisation, so we checked that too'

Thanks for this. We have now ascertained that each analysis is introduced in a clear way, making explicit the rationale of the analysis and how this is linked with our hypotheses. For example (p7 last paragraph):

“To better characterize the mechanisms underlying choice behaviour and to quantify the level of influence exerted by context on value normalization”

And (p9, second paragraph):

“To assess whether our model can explain the main behavioural findings, we used the model and subject-specific parameters estimates to generate simulated data and perform the behavioural analyses on the simulated data.”

b) giving as concise and explicit a statement of the key results as possible, in each of the introduction/discussion/abstract.

Instead of stating 'there was a relationship between x and y' stating the nature of the relationship 'as x increases, y increases'.

For example (but this is a general comment about the whole paper) in the abstract it says:

"When we examined responses to choices common to both contexts, activity in ventral tegmental area/substantia nigra showed a context-sensitivity consistent with adaptive neuronal gain control, an effect correlating with the influence of context on choice, both across and within participants"

why not say:

"Activity in dopaminergic regions (VTA/SN) was context sensitive, such that a reward of a given size evoked a greater response in the context of a low reward block. This effect was stronger in participants who were more prone to gamble in low reward blocks. Within participants, increased HPC activity at the start of each block predicted increased context sensitivity in that block, both behaviourally and in the dopaminergic midbrain signal"

...or some similar explicit statement that reflects your exact conclusions?

Obviously, it is the authors' own work and they should write in their own style. I am only pointing out that I thought a lot of the work of motivating the reader was left up to the reader himself and, had I not been obliged to finish the paper because I was reviewing it, I might have gotten a bit lost in the sea of analyses and given up, which would have been a pity because the results, when stated concisely, are actually quite interesting and important.

Thanks for this suggestion. We agree this is a clearer way to present the results, and we now have attempted to follow this approach throughout the paper. For instance, we follow the reviewer's advice and we now write in the abstract:

"At the beginning of each block (when information about context is provided), hippocampus is activated and this response is enhanced when contextual influence on choice increases. In addition, response to value in ventral tegmental area/substantia nigra (VTA/SN) shows context-sensitivity, an effect enhanced with an increased contextual influence on choice. Finally, greater response in hippocampus at block start is associated with enhanced context sensitivity in VTA/SN."

Reviewer #2 (Remarks to the Author):

The authors have done a good job in responding to comments. The new PPI analysis substantially strengthens the paper.

Many thanks for your very useful comments.